# Probiotic Characteristics of *Streptococcus thermophilus* and *Lactobacillus bulgaricus* as Influenced by New Food Sources

**DOI:** 10.3390/microorganisms11092291

**Published:** 2023-09-12

**Authors:** Ashly Castro, Ricardo S. Aleman, Miguel Tabora, Shirin Kazemzadeh, Leyla K. Pournaki, Roberto Cedillos, Jhunior Marcia, Kayanush Aryana

**Affiliations:** 1Faculty of Technological Sciences, Universidad Nacional de Agricultura, Road to Dulce Nombre de Culmí, Km 215, Barrio El Espino, Catacamas 16201, Honduras; ashlycastro.ramos@gmail.com (A.C.); mtaborafuentes@agcenter.lsu.edu (M.T.); jmarcia@unag.edu.hn (J.M.); 2School of Nutrition and Food Sciences, Louisiana State University Agricultural Center, Baton Rouge, LA 70802, USA; rsantosaleman@lsu.edu (R.S.A.); rcedil1@lsu.edu (R.C.); 3Department of Dairy and Food Science, South Dakota State University, Brookings, SD 57007, USA; shirin.kazemzadehpournaki@sdstate.edu; 4Department of Food Engineering, Near East University, Lefkosa 99150, Cyprus; leyla.kazemzade@neu.edu.tr

**Keywords:** probiotic, *Streptococcus thermophilus*, *Lactobacillus bulgaricus*, *Solanum mammosum*, *Dioon mejiae*, *Amanita caesarea*

## Abstract

The current research aimed to evaluate the potential effects of *Solanum mammosum*, *Dioon mejiae*, and *Amanita caesarea* on *Streptococcus thermophilus* and *Lactobacillus delbrueckii* subsp. *bulgaricus* survival and performance after exposure to different harsh conditions such as bile, acid, gastric juice, and lysozyme to mimic the digestive system from mouth to the intestine. Probiotic protease activity was observed to evaluate the proteolytic system. Probiotics were cultured in a broth mixed with plant material, and after incubation, the results were compared to the control sample. Therefore, plant material’s total phenolic compound, total carotenoid compound, antioxidant activity, sugar profile, and acid profile were obtained to discuss their impact on the survival of probiotics. The results indicate that *Amanita caesarea* negatively affected probiotic survival in the bile tolerance test and positively affected *Lactobacillus bulgaricus* in the protease activity test. Otherwise, the other plant material did not change the results significantly (*p* > 0.05) compared to the control in different tests. Consequently, *Solanum mammosum* and *Dioon mejiae* had no significant effects (*p* > 0.05) in increasing probiotic survival.

## 1. Introduction

The most enriched and functional food with probiotics is dairy products. At the same time, the entire collection of factors involved is yet unknown. Probiotics reveal several encouraging findings and patterns regarding immune regulation [1]. Probiotics provide some advantages regarding the inflammatory responses of localized immunity (by preserving gut wall integrity) and methods to protect (by enhancing non-specific and specific arms of the immune system). Whilst probiotics may be used independently, one of the most practical methods to apply probiotics is to improve valuable food items with them, perhaps together with other ingredients such as dietary fibers, vitamins, and minerals [2]. The variation and structure of intestinal bacteria significantly influence host well-being by controlling the output of metabolic pathways such as supplements and short-chain fatty acids, enzyme formulation, cell-to-cell connections, immune system and neuroendocrine replies, and nutrient metabolism and absorption. As a result, the variety and stability of the intestinal flora population, along with the existence or lack of essential types, are critical for balanced maintenance [3]. The intestinal flora is reported to compromise approximately 100 trillion microorganism species, such as fungi, viruses, bacteria, and eukaryotes [4].

Probiotics separated from foodstuff are more resistant to variations in pH and heat throughout meal preparation. However, they are less resistant to the gastrointestinal environment than probiotic bacteria isolated from the intestine. The Food and Agriculture Organization (FAO) defines probiotics as bacteria that provide significant medical advantages to the host. These bacteria should be able to survive the transit through the intestinal system and increase within the gut. This implies that they should be capable of refilling in the presence of bile and gastric secretions or be absorbed alongside the food material to allow for longevity and transit through the intestine [5]. The gut is undoubtedly the habitat of a steady and “precisely calibrated” ecology of around 1 × 10^11^–10^12^ bacteria [6]. 

Until recently, probiotic meal sources were nearly entirely limited to dairy products. On the other hand, fermented foods of organic origin are gradually being explored as carriers for probiotic culture integration. A basic grasp of the function of vegetable and fruit structures, especially regarding probiotic strain administration, currently needs to be improved. Therefore, the effort is to use new sources, such as vegetables and fruits, for probiotics [7].

*Solanum mammosum* (Chi), a tropical fruit (native to South America) and a member of the Solanaceae family, has long been utilized for medical reasons, even though it has long been claimed that this species contains steroidal alkaloids like solasodine and its compounds [8]. 

*Dioon mejiae* (DM), also named teosinte, is a growing fossil tree discovered in Honduras. It is native to the Department of Olancho, with the largest population observed in Rio Grande and Saguay municipalities. This teosinte is a cycad, a species group that is heavy or bulky before the separation between monocots and dicotyledons, which is utilized including all current grains used for nutrition, and it is not linked to the variety of *Zea mays* also recognized by the common name, teosinte. Seeds of this plant are used to make flour that is prepared for making typical food. The starch content of the sago seed is so high that is used as a nutritional booster [9]. *Amanita caesarea* (AC), popularly recognized as Caesar’s mushroom, is an edible fungus. It features an eye-catching orange crown, yellow gills, and a stem. Its importance has been recognized since the earliest days of the Romans [10]. Mushrooms (Amanita caesarea) are an excellent protein, fiber, and mineral composition source. Furthermore, they contain many phytochemical constituents, including phenolics, ascorbic acid, tocopherols, and carotenoids. As a result, mushroom (*Amanita caesarea*) is a healthful item to incorporate in a regular diet [11]. 

The culture bacterium *Lactobacillus bulgaricus* (LB) and *Streptococcus thermophilus* (ST) is yogurt manufactured by fermenting milk. Lactose is transported into the ST cell from beyond the cell membrane by membrane-bound β-galactoside permease. Lactose is broken down into glucose and galactose by β-galactosidase once within the cell wall. Lactose is a source of power for living organisms. Because bile and lactose impact cell wall integrity in varied contexts, one would wonder if lactose affected the bile tolerance of yogurt culture bacteria. Lactobacilli MRS agar was used to count LB-12, whereas M17 agar was used to count ST-M5 [12]. LB and ST are Gram-positive bacteria which are used as a starter for yogurt fermentation, and are considered probiotics, enhancing metabolism and improving bacterial balance. However, LB has low tolerance against acidic pH and bile [13]. ST has been used to produce fermented beverages such as kefir and viili. The current study aims to observe the synergistic effects of different sources (Chi, Cho, and Teo) on probiotics to increase their resistance in the digestive system from the mouth to the intestine, and various tests were applied to study variables’ effects on probiotics.

## 2. Materials and Methods

### 2.1. Plant Material

The food sources (Cho, Teo, and Chi) were compiled and categorized in the Guapinol Biological Reserve, Marcovia Municipality, Choluteca Department (Honduras), between August and September 2021. Chi *Solanum mammosum*, Teo-*Dioon mejiae*, Cho-*Amanita caesarea* powder were combined with water (10% *wt.*/*wt.*) and then stored at −80 °C temperature and lyophilized (LIOTOP model L 101). The solution was freeze-dried for 48 h at −73 to −76 °C and 0.1 to 0.3 Pa (LIOTOP model L 101). An industrial mill (LABOR model SP31, Spain) was used to grind the lyophilized pulp (LABOR model SP31). Plastic bags were used to vacuum-pack the pulp powder. 

### 2.2. Experimental Design

The food resources (Cho, Teo, and Chi,) were examined for their survivability under different stresses. The viability characteristics and bacterial resistance with the synergistic effects of plant material were examined by applying stress situations, such as acid, bile, gastric juices, lysozyme, and protease activity, applied on ST-M5 and LB-12 and were measured separately (Cho Teo, and Chi, ) using M17 agar and MRS agar at different time points. Three species (Chi, which is fruit; Teo, trees’ seeds; and Cho, a mushroom) were investigated at 2% (*w*/*w*). Different stress tests were applied, including pH 2, oxgall salt (0.3%), lysozyme (100 mg/L), gastric juice (pepsin (0.32%), and NaCl (0.2%) to compare each test, along with a control which was not paired with plant materials. The plate counts were determined to evaluate the viability. All experiments were repeated in triplicate with duplicate readings. 

### 2.3. Bile Tolerance Test

The bile tolerance of ST and LB (Chr. Hansen, Milwaukee, WI, USA) were evaluated by using the Pereira and Gibson (2002) method [14] with slight modification. After the M17 (ST) and MRS (LB) broths were prepared, freeze-dried plant material, water, and 1.5 g bile salt (oxgall salt 0.3%) were added and autoclaved. The culture bacteria were cultured in MRS broth (CriterionTM, Hardy Diagnostics, Santa Maria, CA, USA) for LB and M17 broth (CriterionTM, Hardy Diagnostics, Santa Maria, CA, USA) for ST with 0.2% (*wt*/*v*) sodium thioglycolate (Sigma-Aldrich, St. Louis, MO, USA) and bile salt oxgall . The cultured broths were incubated at 37 °C for ST, and 43 °C for LB. An 11 mL sample was collected at several periods (0, 4, and 8 h), 10-fold diluted in peptone water, and plated in duplicate [14]. 

### 2.4. Acid Tolerance Test

ST and LB’s acid tolerances were evaluated by inoculating the starter cultures at 10% (*v*/*v*) into the acidified M17 and MRS broths acidified with 1 N HCl to pH 2.0. These acidified M17 and MRS broths-containing culture were incubated at a temperature of 37 °C for ST for 24 h and 43 °C for LB for 72 h, respectively. A 1 mL sample was collected at several periods (0, 30, and 60 min) [14].

### 2.5. Protease Activity

ST and LB protease activity were evaluated using the o-phthaldialdehyde (OPA) spectrophotometric test established by Oberg et al. (1991) [15]. After incubation of ST and LB in sterile skim milk helping to maintain them as Shihata and Shah (2000) [16], ST and LB were grown at 43 °C, and 37 °C, respectively, for 24 h, and then 2.5 mL of each sample was combined with 1 mL distilled water and 10 mL of 0.75 N trichloroacetic acid (TCA) to give a final concentration of 7.7%. All samples were filtered in the room using a Whatman Number 2 filter paper for 10 min. A double portion of each TCA filtrate was examined by the OPA spectrophotometric test utilizing spectrophotometry at 340 nm (Nicolet Evolution 100, Thermo Scientific; Madison, WI, USA).

### 2.6. Viability of Culture Bacteria

To analyze the growth of ST and LB, they were evaluated by plate counting. The inoculation of starter cultures 10% (*v*/*v*) was carried out into 19 g MRS agar (Difco, Detroit, MI, USA), added to sodium thioglycolate (1%) for LB and to 19 g M17 Agar (Difco) lactose concentration (0.5% *wt*/*v*) for ST at pH 6.5 and 6.8, respectively. Samples of LB and ST were incubated under anaerobic conditions at 43 °C for 72 h and 37 °C for 16 h, respectively [17].

### 2.7. Tolerance to Simulated Gastric Juice

The tolerance of ST and LB to function substances in synthetic gastric juice (SGJ) was tested using the method described by García-Ruiz et al. (2014) [18] and Liao et al. (2019) [19] with a minor modification. The SGJ was prepared using H_2_O, pepsin 0.32% (Sigma-Aldrich), NaCl 0.2%, NaOH, and HCl for pH adjudication. With 1 M HCl and 1 M NaOH, the simulated gastric juice was modified to five concentration gradients (pH 7.0, 5.0, 4.0, 3.0, and 2.0). Starter cultures of ST and LB were inoculated 10% (*wt*/*v*) into SGJ and incubated for 30 min under anaerobic conditions at 37 °C (ST) and 43 °C (LB). Plates were counted at 0 and 30 min of incubation to determine live bacteria.

### 2.8. Lysozyme Tolerance Test

LB and ST resistance to lysozymes was evaluated according to Zago et al. (2011) [20] with slight modification. The electrolyte solution was used to control the lysozyme tolerance test and to imitate in vivo dispersion by saliva. Bacteria cultures were inoculated (10% *wt*/*v*) to a sterile electrolyte solution (SES) of 0.22 g/L CaCl_2_, 6.2 g/L NaCl, 2.2 g/L KCl, and 1.2 g/L NaHCO_3_ in the presence of lysozymes (100 mg/L) (Sigma-Aldrich). The tests comprised microbial cultures in SES without lysozymes. After incubation, bacterial counting was performed on MRS agar (72 h and 43 °C) and M17 (24 h and 37 °C). The survival expectancy was determined by comparing the CFU/mL after 30 and 120 min to the CFU/mL at 0.

### 2.9. Enumeration of Culutre Bacteria

The components for ST agar are (ST agar; 10.0 g sucrose (Amresco, Solon, OH, USA) tryptone (Becton, Dickinson, and Co., Sparks, MD, USA), 2.0 g K_2_HPO_4_ (Fisher Scientific, Fair Lawn, NJ, USA), and 5.0 g yeast extract that were diluted in 1 L of distilled water. The medium’s pH level was regulated to 6.8 ± 0.1, and 6 mL of 0.5% bromocresol purple (Fisher Scientific, Fair Lawn, NJ, USA) and 12 g of agar were applied. The media was sterilized at 121 °C for 15 min [21]. Samples were diluted using 99 mL of MgCl_2_ and KOH before being transferred to Petri dishes. Petri dishes were incubated on a Petri plate anaerobically at 37 °C for 1 day. After overnight incubation, a colony counter (Darkfield Quebec Colony Counter, American Optical, Buffalo, NY, USA) was applied to aid in the counting of populations [22]. Preparation of the MRS broth of LB, including 1 L of distilled water, was added to 55 g of MRS broth powder (Difco, Becton, Dickinson and Co., Sparks, MD, USA). We utilized 1 N HCl to reduce the pH to 5.2. To thoroughly disperse the particles, this media was boiled under stirring and sterilization at 121 °C for 15 min [23]. Following plating into inoculated medium, MRS broths were pipetted to various formulations using 99 mL of sterilizing phosphate buffer 0.1% (*w*/*v*). After 72 h, these LB plates were heated anaerobically at 43 °C. A Quebec Darkfield Colony Counter was used in calculations (Leica Inc., Buffalo, NY, USA) [24].

### 2.10. Statistical Analysis

Data were analyzed using the General Linear Model (PROC GLM) of the Statistical Analysis Systems (SAS). Differences of least square means were used to determine significant differences at *p* < 0.05 for the main effect (ingredients vs. control). Data are presented as mean ± standard error of means. Significant differences were determined at α = 0.05. 

## 3. Results and Discussion

### Bile, Acid, Lysozyme, and Gastric Juice Effects on Probiotic Resistance and Protease Activity

Figure 1 represents ST and LB tolerance to bile to compare whether plant resources help to maintain probiotics longer in media. Cho caused the highest stability in ST counts (11.41 Log CFU/mL), and there was no significant difference (*p* < 0.05) among the control, Chi, and Teo at the initial. For both bacteria, the ingredient effect and the time effect were significant (*p* < 0.05); whereas the interaction effect (ingredient concentration × time) was not significant (*p* > 0.05). The nipple fruit and teosinte flour did not affect the bile tolerance of both bacteria, while the Caesar mushroom results reported lowered counts when compared to the control, and the growth decreased over time. The growth remained stable until 4 h and decreased significantly (*p* < 0.05) at 8 h.

After 4 h, a significant decrease in ST counts for the Cho treatment (6.03 Log CFU/mL) could be observed compared to the control, Cho, and Chi, which showed no significant difference in ST counts. The Cho treatment continued to decrease (6.09 Log CFU) until 8 h, and the control, Chi, and Teo treatments experienced a decrease (9.14, 9.83, and 10.00 Log CFU/mL, respectively) and showed no meaningful difference. For LB counts, the pattern is different. The control significantly showed the highest counts (9.34 Log CFU/mL). The lowest bacterial counts resulted from the Cho treatment (5.14 Log CFU/mL) at 0, 4, and 8 h of the experiment. Although Chi and Cho had a downward trend the entire time, they were not significantly different from each other after 8 h. The highest LB count at the end of the experiment against bile resulted from Teo treatments (7.45 Log CFU/mL). In general, ST was more stable than LB against bile according to the results.

The acid tolerance of SB and LB is shown in Figure 2. Acid tolerance was examined over 60 min for the bacterial count in order to examine the effects of plant sources on probiotics activity and survival, so as to mimic stomach conditions. For both bacteria, the ingredient effect and the time effect were significant (*p* < 0.05); the interaction effect (ingredient concentration × time) was not significant (*p* > 0.05). The nipple fruit and teosinte flour did not affect the bile tolerance of both bacteria, while the Caesar mushroom analysis reported lowered counts when compared to control. ST counts significantly decreased (*p* > 0.05) in all treatments after 60 min. The Chi treatment had the highest resistance (8.31 Log CFU/mL) against acid, and the Cho treatment had the lowest survival (6.38 Log CFU/mL) after 60 min. All treatments indicated no significant difference from each other by the end of the experiment. LB counts for all treatments (control, Cho, Chi, and Teo ) differed from the ST results. The control always experienced the lowest bacterial counts (5.70, 5.78, and 4.35 Log CFU/mL, respectively). The Cho treatments showed no significant changes after 30 min, and the Chi treatment had the highest bacterial counts (7.24 Log CFU/mL) at first, which showed a significant decrease (*p* < 0.05) until 30 min. The addition of the plant sources was ineffective on the acid tolerance of bacteria after 30 min. LB was sensitive to acid, was not detected after 30 min, and had a lower bacterial count than ST.

The protease activity of ST and LB is shown in Figure 3. The observation of protease activity of ST showed an increase in 12 h and 24 h for all treatments. For both bacteria, the ingredient effect and the time effect were significant (*p* < 0.05); nevertheless, the interaction effect (ingredient concentration × time) was not significant (*p* > 0.05). The control, which showed no significant difference (*p* > 0.05) from Teo, had the lowest absorbance (1.399 µL/mol), and Cho, which showed no significant difference from Chi, had the highest absorbance (1.926 µL/mol) at the end of the experiment. The Chi and Cho treatments showed no significant difference from 12 to 24 h. LB protease activity was the lowest for the control treatment at the time 0 h, decreased over 24 h, and was the lowest protease activity (452 µL/mol) among others. Chi was the only treatment that was the highest significantly (901.33 µL/mol) at time 0 h; first, it increased after 12 h and then increased slightly after 24 h. The lowest protease activity (452 µL/mol) for LB was detected in the control treatment after 24 h. LB’s protease activity was higher than ST’s at the end of the experiment.

ST and LB’s resistance to lysozymes is shown in Figure 4. For *L. bulgaricus*, the ingredient effect and the time effect were significant (*p* < 0.05); nevertheless, was not significant (*p* > 0.05). For *S. thermophilus*, the ingredient effect, the time effect, and the interaction effect (ingredient concentration × time) were significant (*p* < 0.05). The initial count of ST for the control was 10.46 Log CFU/mL, which showed no significant difference from other treatments. The highest ST count (11.37 Log CFU/L) resulted from the control sample and Chi had the lowest ST count (10.51 Log CFU/mL) at the end of the lysozyme resistance experiment. The Teo treatment experienced an initial decrease after 1 h and then it increased after 2 h. Chi experienced a dramatic increase after 1 h and then decreased after 2 h to 10.51 Log CFU/mL. The LB bacterial count experienced a decrease in all treatments. The treatments indicated no significant difference at the initial count, and they decreased after 2 h. The lowest LB counts (which resulted from the control treatment at final (5.03 Log CFU/mL) were not significantly different (*p* > 0.05) from the Cho and Teo treatments. The other treatments (control, Cho, and Teo) indicated the same LB counts at the end of the experiment.

ST and LB’s resistance to stimulated gastric juice is shown in Figure 5. For both bacteria, the ingredient effect and the time effect were significant (*p* < 0.05), whereas the interaction effect (ingredient concentration × time) was not significant (*p* > 0.05). Bacterial counts of ST for control, Chi, and Teo showed the same pattern from pH 2 to 7; they were increased slightly without significant difference from each other. The control indicated the highest bacterial counts for LB, and Cho always showed the lowest ST counts. The LB count for the control treatment was highest at pH 7 and the lowest result was observed from Cho treatments.

## 4. Discussion

Chi belongs to the Solanaceae family and is rich in polyphenolic compounds containing antioxidants, anticancer, anti-pest, antimalarial, and biotransformed compounds. According to the results from Pilaquinga et al. (2021) [25], Chi contains gallic acid, kaempferol, catechin, resveratrol, and quercetin, which scavenge oxygen-derived free radicals [26]. Chi fruit was conventionally used for respiratory disease, prohibiting pyocyanin production, and Gram-positive Pseudomonas aeruginosa film formation. For example, Chi comprises *solasodine*, a poisonous tetratogenic alkaloidal chemical with antifungal and antibacterial effects [27]. Chi treatments comprise high TPC, TCC, and antioxidant activity content, which might be the reason for the impact on bacterial growth in the protease activity test and LB count in resistance to the lysozyme test. It was discovered that polyphenols and pigments significantly modulate probiotic bacteria, helping them grow and survive [28].

Teo is traditionally used for food and beverages that indicate high polyphenol content. According to El Gendy et al. (2022) [29], it has 83.00 ± 4.1% inhibition in the ABTS method and lower TPC (37.14 μg GAE/mg) compared to the obtained results. Teo is rich in nutritional values and calories due to carbohydrates such as starch with high amylopectin, essential amino acids, dietary fiber, and high unsaturated fatty acids. Glucose and fructose were reported at 0.12 ± 0.01 g/100 g and 0.24 ± 0.01 g/100 g, respectively, [9] in Teo, while glucose content in the current study was higher (0.27 ± 0.03 g/100 g), and fructose content was lower (0.09 ± 0.02 g/100 g). The high Teo glucose content and the highest TCC content might be the cause of the highest bacterial count [30] after a significant decrease at 4 h and increase until 8 h in the bile resistance test. Glucose accelerates bacterial growth, and acid and carotenoid production stimulate bacterial growth by providing essential amino acids and hypoxanthine [31]. Additionally, Teo contains a high amount of tartaric acid, L-malic acid, and succinic acid, which provide an acidic environment. According to Donkor et al. (2006) [32], an acidic condition in yogurt culture increases *Lactobacillus acidophilus* L10 and *Lactobacillus paracasei* L26 survival. Microencapsulation techniques are recently used strategies to improve the viability of probiotics in dairy probiotic products. Alginate–milk microspheres encapsulate *L. bulgaricus* to increase survivability [33].

Cho showed 89.74% radical scavenging ability at 6.0 mg/mL of concentration, according to Zhu et al. (2016) [34], while at 0.5 mg/mL concentration, 14.43 ± 1.02% antioxidant activity was observed. The antioxidant ability of Cho might be due to flavonoids, tannins, and phenolic acids [35]. Cho monosaccharides are mainly D-glucose, and α-D-lyxose, and the backbone was made from 1,4 linked α-D-glucose and 1,3,6-linked α-D-glucose having branches from 1-linked α-D-lyxose [34]. The Cho treatment’s effects on bile and acid resistance were not comparable with other treatments’ effects on probiotic growth, suggesting that Cho might negatively affect probiotic resistance. The Cho results reported higher protease activity compared to other treatments. The proteolytic system is essential for probiotic bacteria, especially lactic acid bacteria, to develop flavor in fermented products. Bacteria metabolize protein to use nitrogen [36].

## 5. Conclusions

The addition of the plant sources (Cho, Chi, and Teo) helped to improve ST and LB growth and performance in harsh conditions like acid, lysozyme, bile, and gastric juice. Plant sources of Chi and Teo did not negatively affect ST and LB viable counts, while Cho harmed bile tolerance and increased the protease activity in probiotics.

## Figures and Tables

**Figure 1 microorganisms-11-02291-f001:**
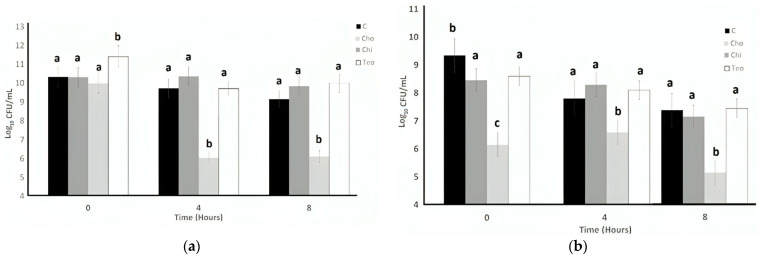
Bile tolerance of *S. thermophilus* (**a**) and *L. bulgaricus* (**b**) as influenced by treatments over 8 h. Average of three replicates. Chi = *Solanum mammosum*, Teo = *Dioon mejiae*, and Cho = *Amanita caesarea*, and C = control. ^a,b,c^ Different letters indicate significant difference among treatments with a time point (*p* < 0.05).

**Figure 2 microorganisms-11-02291-f002:**
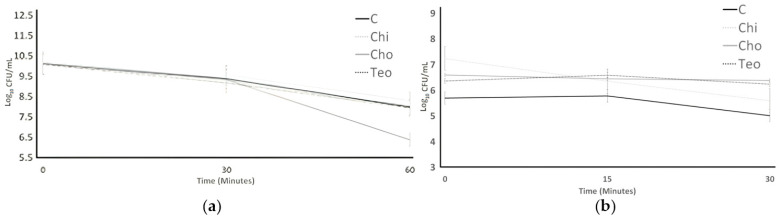
Acid tolerance of ST (**a**) and LB (**b**) as influenced by treatments over 60 min. Average of three replicates. Error bars represent SE. Chi = *Solanum mammosum*, Teo = *Dioon mejiae*, and Cho = *Amanita caesarea*, and C = control.

**Figure 3 microorganisms-11-02291-f003:**
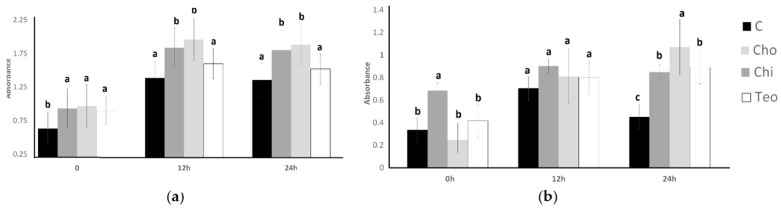
The protease activity of ST (**a**) and LB (**b**) was influenced by different treatments over an incubation period of 24 h. Average of three replicates. Chi = *Solanum mammosum*, Teo = *Dioon mejiae*, and Cho = *Amanita caesarea*, and C = control, ^a,b,c^ Different letters indicate significant difference among treatments with a time point (*p* < 0.05).

**Figure 4 microorganisms-11-02291-f004:**
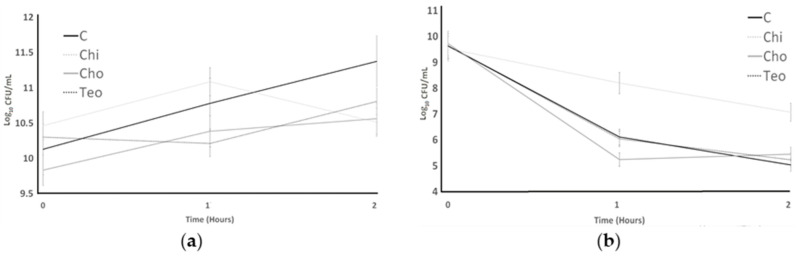
Resistance to lysozymes of ST (**a**) and LB (**b**) as influenced by treatments during incubation time of 2 h. Average of three replicates. Error bars represent SE. Chi = *Solanum mammosum*, Teo = *Dioon mejiae*, and Cho = *Amanita caesarea*, and C = control.

**Figure 5 microorganisms-11-02291-f005:**
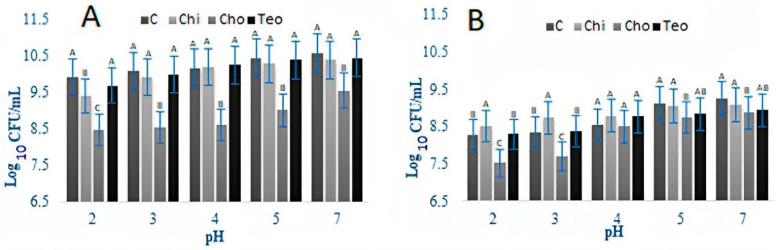
Resistance to simulated gastric juice of ST (**A**) and LB (**B**) as influenced by treatments over different pH conditions. Chi = *Solanum mammosum*, Teo = *Dioon mejiae*, and Cho = *Amanita caesarea*, and C = control. ^A,B,C^ Different letters indicate significant difference among treatments with a pH measurement.

## Data Availability

Not applicable.

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
