# Peer review of "Probiotic Characteristics of *Streptococcus thermophilus* and *Lactobacillus bulgaricus* as Influenced by New Food Sources"

_microorganisms, 2023, doi:10.3390/microorganisms11092291_

Round 1
Reviewer 1 Report
In manuscript "Probiotic characteristics of Streptococcus thermophilus and Lactobacillus bulgaricus as influenced by new food sources" examine the influence of natural substances on the growth and survival of potentially probiotic strains. Have the mentioned strains already proven probiotic action or do they have potential probiotic action? Namely, if we are talking about starter cultures, this must be clearly emphasized.
At the first mention of the bacterium Lactobacillus bulgaricus, write its full Latin name.
Highlight abbreviations at the first mention of natural substances that have been tested.
The introduction is too broad and contains facts that are not necessary to follow the research.
2.9. and 2.10. refer to the determination of the number of bacteria in all previously mentioned experiments? if yes, then they should be added there. It should not be mentioned separately.
It is common to write log10CFU/mL, so change it in the graphs.
Figure 2 could have been displayed as a bar graph and then displayed statistical significance.
Figure 5. Check the standard deviation.
Do plant sources affect better growth as prebiotics? I don't know if the term synergistic effect (in conclusion) is the correct term that should be used.
Author Response
Thank you very much for your comments. Please see the responses attached.

Reviewer 2 Report
The topic of the study is very interesting as it aims to expand the number of probiotics currently in use through the use of completely new products.
However there are some points to consider:
• Wouldn't it be appropriate, before carrying out the researchtests, to carry out a characterization of the three matricesused (Solanum mammosum, Dioon mejiae, Amanita caesarea)? • In figure 1, in line 221, the letters 'A' and 'B' must be insertedin lower case, as shown above. • Moreover, the article should proofread by a native speaker to improve the English language.
Author Response

(The authors gave the same response as above.)

Reviewer 3 Report
Dear Editor,
The original paper entitled “Probiotic characteristics of Streptococcus thermophilus and Lactobacillus bulgaricus as influenced by new food sources” is appropriately well-written, designed, and structured by Castro et al. in suitable English with a clear structure. They evaluated the potential effects of some fruits and plants including Solanum mammosum, Dioonmejiae, and Amanita caesarea on some probiotic lactic acid bacteria including Streptococcus thermophilus and Lactobacillus bulgaricus survival and their performance after exposure to different environmental conditions such as the presence of bile, acid, gastric juice, and lysozyme to mimic the human digestive system. This study is very interesting and practical. However, there are some main concerns that should be addressed by the authors,
- Microencapsulation techniques are the recently used strategies to improve the viability of probiotics in non-dairy probiotic products such as probiotic fruit juices. You should address these techniques and discuss how they can be considered as alternatives and compare the results with some studies regarding using microencapsulation for probiotic fruit juice production using the same organisms.
- Streptococcus thermophilus and Lactobacillus bulgaricus are very strong organisms and resistant probiotic bacteria against harsh conditions. Why the authors did not evaluate sensitive and susceptible probiotic bacteria such as bifidobacterial strains in this study to compare the results accordingly? If it is possible, I highly recommend evaluating, if not, please explain your reasons and discuss it.
- Discussion section is very limited for a normal article. Please develop the discussion section appropriately. There are several scientific points of view in this manuscript which can be discussed comprehensively. Also, there is a results and discussion section but the authors did not discuss anything in this section.
- The quality of the figures is not suitable (figure 4). Also, please make the figure more professional which is suitable for a high-impact factor and prestigious journal such as the Microorganisms journal.
Author Response

(The authors gave the same response as above.)

Round 2
Reviewer 1 Report
No further comments!
Reviewer 3 Report
Dear Editor,
I have no more comments and all revisions have been addressed successfully.